# Confidence and knowledge in emergency management among medical students across Colombia: A role for the WHO basic emergency care course

**Katelyn Moretti**[1,2]*, **Adam R. Aluisio**[1,2], **Benjamin Gallo Marin**[1], **Chuan-Jay Jeffrey Chen**[3], **Catalina González Marqués**[1,2], **Francesca L. Beaudoin**[1,2], **Melissa Clark**[4], **Andrés Patiño**[5], **Heidy Carranza**[6], **Andres Duarte**[6], **Atilio Moreno**[6], **Leonar G. Aguiar**[6], **Christian Arbelaez**[1,2,6]

**1** Warren Alpert Medical School of Brown University, Providence, Rhode Island, United States of America, **2** Brown Emergency Medicine, Providence, Rhode Island, United States of America, **3** Massachusetts General Hospital, Boston, Massachusetts, United States of America, **4** Brown School of Public Health, Providence, Rhode Island, United States of America, **5** Department of Emergency Medicine, Emory University School of Medicine, Atlanta, Georgia, United States of America, **6** Pontificia Universidad Javeriana y Hospital Universitario San Ignacio, Bogotá, Colombia

* katelyn_moretti@brown.edu

## Abstract

### Introduction

Globally, medical students have demonstrated knowledge gaps in emergency care and acute stabilization. In Colombia, new graduates provide care for vulnerable populations. The World Health Organization (WHO) Basic Emergency Care (BEC) course trains frontline providers with limited resources in the management of acute illness and injury. While this course may serve medical students as adjunct to current curriculum, its utility in this learner group has not been investigated. This study performs a baseline assessment of knowledge and confidence in emergency management taught in the BEC amongst medical students in Colombia.

### Methods

A validated, cross-sectional survey assessing knowledge and confidence of emergency care congruent with BEC content was electronically administered to graduating medical students across Colombia. Knowledge was evaluated via 15 multiple choice questions and confidence via 13 questions using 100 mm visual analog scales. Mean knowledge and confidence scores were compared across demographics, geography and prior training using Chi-Squared or one-way ANOVA analyses.

### Results

Data were gathered from 468 graduating medical students at 36 institutions. The mean knowledge score was 59.9% ± 23% (95% CI 57.8–62.0%); the mean confidence score was

**Data Availability Statement:** All relevant data are within the paper and its Supporting information files.

**Funding:** This research was funded, in part, through the Brown Emergency Medicine Innovation's grant through Brown Emergency Physicians, Inc. (KM). The funders did not play a role in the study design, data collection and analysis, decision to publish, or preparation of the manuscript. https://brownphysicians.org/brown-emergency-medicine/.

**Competing interests:** The authors have declared that no competing interests exist.

59.6 mm ±16.7 mm (95% CI 58.1–61.2). Increasing knowledge and confidence scores were associated with prior completion of emergency management training courses (p<0.0001).

## Conclusion

Knowledge and confidence levels of emergency care management for graduating medical students across Colombia demonstrated room for additional, specialized training. Higher scores were seen in groups that had completed emergency care courses. Implementation of the BEC as an adjunct to current curriculum may serve a valuable addition.

## Introduction

Globally, newly graduated physicians have demonstrated knowledge gaps in emergency care. For example, in the United Kingdom, only 55% of interns adequately performed cardiopulmonary resuscitation [1]. Similarly, in India, first year physicians who were tested on advanced life support answered questions correctly only 50% of the time [2]. Understanding the knowledge gaps of recently graduated physicians in a specific country or region may be helpful in identifying educational targets for supplemental training. In the South American country of Colombia, addressing common gaps in emergency care knowledge early after graduation is of particular importance. First year physicians are frequently assigned to one year of social service during which they care for vulnerable populations, often in resource-limited settings and potentially with limited oversight [3, 4]. Thus, these practitioners need to be proficient in acute medical care.

Emergency medicine training has been growing in emphasis across Latin America and specifically, in Colombia. This growth is reflected in Colombian graduate medical education with the creation of emergency medicine residencies [5] as well as in undergraduate medical education with the development of specific emergency care curricular guidance from the Ministry of Education and the *Asociación Colombiana de Facultades de Medicina* (ASCOFAME). However, undergraduate emergency medicine curriculum is not standardized nationally. Because new physicians across the world show knowledge gaps in emergency care, it is likely that these professionals in Colombia would benefit from the national implementation of a standardized and economically accessible post-graduate curriculum, particularly if such program centers on the specific needs of these providers.

The Basic Emergency Care course (BEC) developed by the World Health Organization (WHO) is an open-access curriculum that covers emergency care across five content areas—the initial approach to resuscitation, dyspnea, trauma, shock, and altered mental status—and is tailored to setting with resource barriers [6, 7]. In 2015, the WHO successfully piloted the BEC course with frontline providers in Uganda, United Republic of Tanzania, and Zambia [8]. Post-test knowledge scores and confidence levels were significantly improved after implementation [6, 9]. Recently, the WHO completed a Spanish translation of the course.

The BEC is delivered as five modules (excluding an Introduction module). A Participant Workbook is given to learners, which includes the content of the course along with exercises that include free-response case-based questions and four-option multiple choice questions. The Participant Workbook also includes a Skills Station section with a checklist-based approached to the skills learners must demonstrate. In addition, supplemental mobile applications that provide interactive case-based training on trauma ABCDEs (Airway, Breathing, Circulation, Disability, Exposure) are also available. Instructors are provided with an annotated

Facilitator Guide along with digital slides for each module that follow the sequence of the Participant Workbook. The course is both free and open-access and can be accessed through the WHO website [7].

The BEC has never been implemented specifically for medical students nor in Latin America; therefore, its pertinence and associated impacts for this learner group is unclear. In addition, the specific educational needs related to emergency care amongst recently graduated physicians in Colombia have not been described. This study assessed the baseline knowledge and confidence in emergency care that is specifically taught in BEC curriculum amongst graduating medical students to answer the question: does the BEC offer an appropriate supplemental curriculum for medical students across Colombia.

## Methods

### Study and sample design

This cross-sectional study was conducted among Colombian medical students in their last year of medical school and approved through the institutional review board at Pontifical Javeriana University and the Hospital Universitario San Ignacio (FM-CIE-0279-19, 05/13/2019). A quantitative online survey was administered to measure medical student confidence in, and knowledge of emergency care. Medical school participation was recruited through three organizations: The *Asociación Colombiana de Facultades de Medicina* (ASCOFAME, the medical school dean's council), the *Asociación de Sociedades Científicas de Estudiantes de Medicine de Colombia* (ASEMCOL, a Colombian medical student association), and the *Asociación Colombiana de Especialistas en Medicina de Urgencias y Emergencias* (ACEM, the emergency medicine physician association). Survey links were distributed to senior medical students through these intermediates consisting of medical school deans, emergency medicine faculty or medical student ASEMCOL representatives.

### Questionnaire design

The survey was written in Spanish and designed to assess students' baseline knowledge and confidence in the BEC content domains. While the course was open-access at the time of this study, formal knowledge assessments were not yet available. While there were some review questions as the end of course sections, these were not enough for a full evaluation. Therefore, a new evaluation tool, with questions targeted to a similar knowledge level as those provided in the course, was created. The survey was reviewed by Colombian study personnel to ensure language and cultural appropriateness. Confidence and knowledge evaluations were divided into topics covered in the course which included: initial approach to resuscitation, dyspnea, trauma, shock and altered mental status. Additional demographic data collected included gender, age, nationality, and medical school. Factors that could potentially impact students' confidence or knowledge of emergency care were collected including potential past careers, previous trainings, and emergency medicine (EM) rotations during their clinical years.

Confidence level was assessed via 13 questions using 100 mm visual analog scales. Questions addressed confidence specific to managing patients with dyspnea, shock, trauma, altered mental status (AMS), with three items per category for a total of 12 questions. A final question asked for confidence in the management of a critical patient in general. Visual analogue scales were selected as they have previously been validated for the measurement of self-efficacy in resuscitation [10]. Knowledge was assessed within the topics of the initial approach to resuscitation, dyspnea, shock, trauma and AMS via 15 multiple-choice questions (MCQs) with 3 questions for each category. Questions were written by investigators mirroring difficulty levels of the pre- and post BEC test.

The BEC course uses low technology simulation to complete task training. Self-reported number of times previously completing key emergency management skills, specifically taught in the BEC, were collected. These included: securing an airway, treatment of tension pneumothorax, and wound packing for hemorrhage control.

## Tool development, piloting and validation

Survey questions were reviewed by content experts consisting of two Board certified emergency medicine specialists based in the United States with extensive experience in global health training and research, two Colombian emergency medicine specialists, one Colombian epidemiologist and one Colombian internal medicine physician using a modified Delphi technique until content consensus was reached. The survey was then piloted with 18 third-year medical students at Pontificia Universidad Javeriana in Bogotá, Colombia. Responses to each knowledge questions were reviewed for reproducibility. For items with outlying response distributions, questions and answers were re-reviewed for fairness and accuracy.

Internal consistency reliability was assessed for the confidence scale with Cronbach alpha amongst a cohort of pilot students.

## Data collection

Surveys were administered via REDCap™ (Research Electronic Data Capture) digital software with data stored on a password protected database [11]. Initial digital invitations for study participation were sent on May 15, 2019 with a follow up sent on June 11, 2019. The survey remained accessible to all potential participants for three months until August 15, 2019.

## Outcomes

The primary outcome measure for this study was assessment of the mean baseline knowledge in the population of interest. Knowledge level was defined as a percent correct score from the 15 MCQs described above. Confidence level was defined as the average 100 mm visual analogue scale from the 13 confidence questions. Items in both the knowledge and confidence section may have been missing based on participants stopping the survey early (failure to complete) or, because the participant did not know the answer and skipped that specific question (failure to respond). Thus, completion of the last question of the knowledge or confidence section was analyzed to distinguish between failure to complete versus failure to respond. Based on this standardized approach, a complete knowledge score was defined as 14 or more questions answered and a complete confidence score was defined as greater than 10 items answered (S1 and S2 Tables in S2 File).

Previous literature has demonstrated a minimum number of task completions to gain basic confidence as 5 [12]. Thus, medical task responses were categorized as: 0, 1, 2–5 times and greater than 5 times. Confidence levels for these specific tasks were also collected.

## Data analysis

From the overall population size of 4,166 graduating medical students, a required sample size of 352 participants was calculated based on a 5% binomial confidence interval and an assumed population mean proportion of 50% to maintain the most conservative sampling estimate [13].

Data were analyzed using STATA/SE 15.1 statistical software. Descriptive analysis was performed describing participant variables using frequencies with percentages. Overall mean percent knowledge and mean percent confidence scores were calculated with 95% confidence

intervals. Mean knowledge and confidence scores for each subject topic were also calculated with 95% confidence intervals.

Except for length of time for survey completion, data were examined graphically and found to be normally distributed. Parametric comparative assessments were used (two-tailed t-test and ANOVA) to evaluate for significant differences between stratified groups of interest. The Benjamini-Hochberg method for controlling for the false discovery rate [14] was used to determine the statistical significance level of P ≤0.014. The method was applied to P values for comparisons of mean knowledge scores and again for mean confidence scores. The more conservative P value was then selected and used consistently for all values.

Chi-Squared or one-way ANOVA analysis was performed for mean percent knowledge score and mean confidence score stratified by covariates. Mean overall knowledge was compared to the *a priori* maximum score of 75% with a one sample test of proportions. Knowledge scores and confidence scores for students who completed previous emergency care courses were compared to the overall knowledge score or confidence score of the study sample using two-sided T-Test.

Participants removed from analysis via case deletion were compared to participants in the final cohort across variables: age, sex, region, number of training courses, and number of times performing a task via Chi-Squared analysis.

## Results

### Pilot survey

The initial Cronbach alpha demonstrated low reliability (alpha = 0.6) amongst the cohort of pilot students in their responses to confidence questions. Questions were edited prior to data collection in the research population, resulting in an improved Cronbach alpha (alpha = 0.9) amongst the study cohort.

### Demographics

Based on reporting metrics from the *Sistema Nacional de Información de la Educación Superior* (SNIES) database created by the Colombian Ministry of Education [15], there are 6,429 senior medical students in Colombia across 57 medical schools. Invitations to participate were successfully sent to 65% of the Colombian medical student body with invitation emails sent to 54 of the 57 total Colombian public and private institutions. ASCOFAME distributed invitation emails to the university administration of 46 schools; ASEMCOL sent invitations to an additional nine institutions. Lastly, ACEM sent invitations to faculty serving in the major teaching hospitals located in the four main cities in Colombia.

Given the reliance on intermediates for survey distribution, the study population was defined as the student population from medical schools with at least one student accessing the survey. This included the student body of 36 medical schools across Colombia encompassing 4,166 students. Of this initial sample, 714 students consented to participate; 468 completed the knowledge and confidence aspects of the survey and were included in analysis for a response rate of 11.2% (Fig 1).

The median time for completion was 17.5 minutes (IQR 13.1–24.0 minutes). The study sample was predominantly female (62.4%) and between the ages of 19–24 years (69.7%). The majority attended medical school in the Andes region of Colombia (66.5%) followed by Caribe (18.4%) (Table 1). Participants removed via case deletion were not statistically different from the study sample by age, sex, region (S4 Table in S2 File).

Medical schools in Colombia may choose to apply for accreditation through the Ministry of Education, meeting several metrics demonstrating excellence in education [16]. The majority

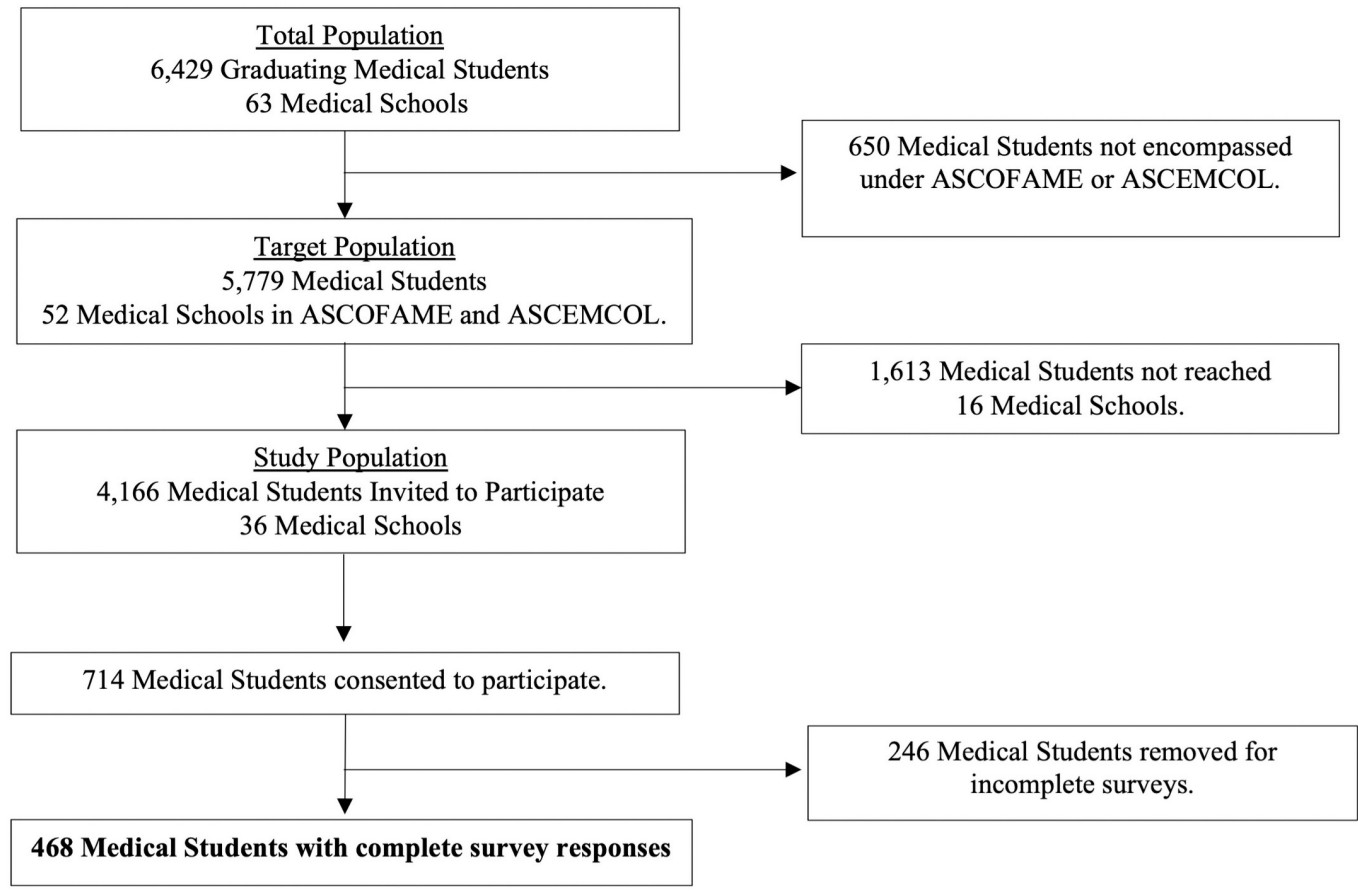

**Fig 1. Study sample.**

of students (62%) attended accredited medical schools. In addition, the majority of students (88.5%) had completed an EM rotation, most commonly greater than four weeks in length (47.2%). The majority of students (81.6%) had completed a standardized resuscitation course with 45.3% completing between 1 and 3 courses and 36.3% completing 4 to 7 courses (Table 2).

The most commonly completed course was the American Heart Association (AHA) Basic Life Support (BLS, 62.2%) followed by the AHA Advanced Cardiovascular Life Support (ACLS, 51.7%, S3 Table in S2 File). Students who had completed BLS, ACLS, Pediatric Advanced Life Support, Neonatal Advanced Life Support or Atención Integral a las

**Table 1. Colombian regions with number and percent study participation.**

| Region | Number of Participants | Percent Participation |
|---|---|---|
| Andes | 311 | 66.5% |
| Caribe | 86 | 18.4% |
| Orinoquía | 38 | 8.1% |
| Pacífico | 31 | 6.6% |
| Amazonas | 0 | 0.0% |
| Caribe | 0 | 0.0% |

**Table 2. Baseline characteristics, knowledge scores and confidence scores in emergency care among 4[th] year medical students in Colombia compared with one-way ANOVA.**

| | # (%) | Mean Knowledge Score (95% CI) | P value | Mean Confidence Score (95% CI) | P Value |
|---|---|---|---|---|---|
| **Overall** | 468 | 59.9 (57.8–62.0) | | 59.6 (58.1–61.2) | |
| **Age** | | | | | |
| 19–24 | 326 (69.7) | 63.3 (60.9–65.8) | <0.0001 | 58.5 (56.74–60.3) | 0.068 |
| 25–44 | 130 (27.8) | 52.8 (48.5–57.2) | | 61.7 (57.9–61.0) | |
| *Missing* | *12 (2.6)* | | | | |
| **Sex** | | | | | |
| Male | 174 (37.2) | 58.9 (55.4–62.4) | 0.45 | 62.8 (60.2–65.3) | 0.002 |
| Female | 292 (62.4) | 60.6 (57.9–63.2) | | 57.7 (55.8–59.6) | |
| *Missing/Other* | *2 (0.42)* | | | | |
| **Region** | | | | | |
| Pacífico | 31 (6.6) | 64.9 (58.8–71.1) | <0.0001 | 55.6 (47.5–63.7) | 0.1637 |
| Caribe | 86 (18.4) | 53.6 (49.6–57.7) | | 57.6 (53.6–61.5) | |
| Andes | 311 (66.5) | 65.0 (62.6–67.5) | | 60.8 (59.1–62.6) | |
| Orinoquía | 38 (8.1) | 27.0 (20.8–33.2) | | 58.0 (51.9–64.2) | |
| *Missing* | *2 (0.4)* | | | | |
| **Accreditation Status** | | | | | |
| Accredited Medical School | 144 (38.0) | 66.0 (63.6–68.4) | 0.002 | 62.7 (60.1–65.4) | 0.007 |
| Not Accredited | 235 (62.0) | 70.8 (68.8–72.7) | | 58.0 (55.8–60.1) | |
| **Number of Training Courses** | | | | | |
| 0 | 86 (18.4) | 48.3 ± 0.22 (43.6–54.0) | <0.0001 | 49.9 (46.1–53.7) | <0.0001 |
| 1 to 3 | 212 (45.3) | 57.4 (54.1–60.7) | | 60.6 (58.5–62.6) | |
| 4 to 7 | 170 (36.3) | 68.8 (65.9–71.7) | | 63.4 (61.1–65.6) | |
| **EM Rotation** | | | | | |
| None/"I don't know" | 54 (11.5) | 52.6 (45.7–59.5) | 0.02 | 50.2 (44.8–55.6) | 0.014 |
| 1–2 weeks | 40 (8.6) | 54 (45.4–62.6) | | 58.7 (54.3–63.2) | |
| 3–4 weeks | 153 (32.7) | 58.8 (54.9–62.7) | | 58.6 (55.8–61.3) | |
| >4 weeks | 221 (47.2) | 63.4 (60.7–66.2) | | 62.9 (60.9–64.8) | |

Enfermedades Prevalentes en la Infancia scored higher on the knowledge assessment as compared to the overall cohort. Those that had completed BLS or ACLS also demonstrated higher confidence levels (S3 Table in S2 File).

## Knowledge and confidence

The mean overall knowledge score was 59.9% ± 23% (95% CI 57.8–62.0%) and mean overall confidence score was 59.6 mm ±16.7 mm out of a 100 mm scale (95% CI 58.1–61.2 mm). A 75% score was defined, *a priori*, as the maximum score for course utility below which, potential knowledge gains would warrant implementation. This is consistent with the defined passing score of the knowledge test provided in the BEC [6]. Mean overall knowledge was statistically different from the *a priori* maximum score of 75% (P = 0.007). Younger age, an increasing number of training courses completed, and attendance at an accredited school were all associated with statistically significantly higher knowledge scores (Table 2) although all were statistically lower than the 75% maximum score.

Male sex, attendance at a non-accredited school and completion of an emergency medicine rotation were associated with a statistically significant increase in confidence levels. In addition, the number of courses completed in emergency management was associated with

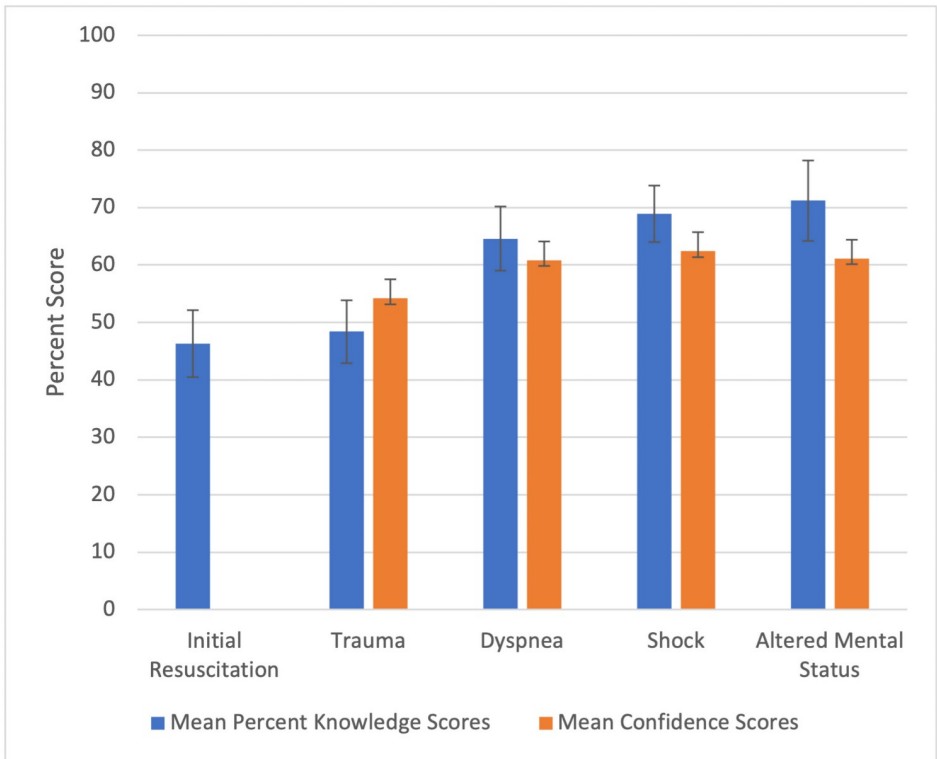

**Fig 2. Mean percent knowledge sores and confidence scores amongst graduating medical students across Colombia by subject areas taught in the WHO basic emergency care course.** *Confidence not measured for Initial Resuscitation.

increased confidence, with increasing number of courses associated with an increasing level of confidence (Table 2, P = <0.0001).

Differences in knowledge scores were found between geographic regions (P<0.0001) with highest scores for students at medical schools in the Andes region (65.0%, 95% CI 62.6–67.5%) and lowest for students in Orinoquía region (27.0%, 95% CI 20.8–33.2%) (Table 2). However, no difference in confidence scores was found between geographic locations (P = 0.16). Knowledge scores were lowest for the initial approach to resuscitation (46.3%, 95%CI 43.4–49.2%) (Fig 2) and highest for altered mental status (71.2%, 95%CI 67.7–74.7%). Confidence was lowest for trauma (54.2 mm 95%CI 52.2–56.2 mm) and highest for shock (62.4 mm, 95%CI 60.7–64.1 mm).

## Skill performance and skill confidence

Twenty eight percent of the study sample reported they had never secured an airway. In addition, the majority had not placed a thoracostomy tube (76.1%) and 48.5% had not treated external hemorrhage. An increasing number of task completions for securing an airway, chest tube placement and external hemorrhage control was correlated with increased confidence in that specific task (Table 3).

## Discussion

Overall, this study demonstrates a level of knowledge and confidence in the management of emergent conditions amongst graduating medical students in Colombia appropriate for the

**Table 3. Number of times graduating medical students reported completing an emergency management skill during training and their associated confidence level in that skill compared with one-way ANOVA.**

|  | # (%) | Mean Confidence in each Skill (95% CI) | P Value |
|---|---|---|---|
| **Number of task performances** | | | |
| _____*Securing an airway* | | | |
| 0 | 131 (28.0) | 47.6 (43.0–52.3) | <0.001 |
| 1 to 5 | 239 (51.1) | 71.3 (68.9–73.7) | |
| >5 | 96 (20.6) | 83.8 (80.8–86.8) | |
| *Missing* | *2 (0.4)* | | |
| _____*Chest Tube Placement* | | | |
| 0 | 356 (76.1) | 45.0 (42.1–48.0) | <0.001 |
| 1 to 5 | 99 (21.2) | 71.6 (67.0–76.2) | |
| >5 | 10 (2.14) | 82.9 (72.2–93.6) | |
| *Missing* | *3 (0.64)* | | |
| _____*External hemorrhage control* | | | |
| 0 | 227 (48.5) | 44.0 (40.6–47.4) | <0.001 |
| 1 to 5 | 217 (46.4) | 40.5 (35.6–45.3) | |
| >5 | 23 (4.9) | 80.0 (74.0–86.0) | |
| *Missing* | *1 (0.2)* | | |

future implementation of the BEC. Results suggest this course could provide potential adjunct support to the current emergency care curriculum. Regardless of medical school accreditation, location, student gender, or previous courses, there was no individual group analyzed in this study that had a mean score above the *a priori* maximum score of 75%. In fact, knowledge scores were similar to the pretest scores of previous BEC learner groups which included front-line providers such as medical officers and nurses from Tanzania, Uganda and Zambia [5]. Similarly, confidence scores for the management of patients needing emergent care for various life-threatening situations were low, suggesting an opportunity for curricular enhancements.

As of the date of submission, this is the first attempt to identify medical students as a learner group for the WHO, besides one study that included 2 medical students in Nigeria [17]. Moreover, of all the studies examining emergency care in LMICs, this is the first to break down results by provider gender, age, and number of previous training courses. Interestingly, a number of trends did emerge from the results. Confidence in providing emergency care was higher among male than female students, correlating with studies in other settings that show male medical students to appear more confident than female students in clinical encounters [18]. Meanwhile, there was no statistical difference in knowledge scores between genders. Similar to other studies, both confidence and knowledge increased with the number of training courses that students had previously taken [1, 2]. This suggests that, similar to other settings [6, 19, 20], implementation of the BEC at the medical school level could increase the knowledge and confidence of medical students who will go on to provide emergency medical care to the vulnerable, underserved, often rural populations during their year of social service [6, 19, 20].

In addition, emergency medicine training in medical school is not standardized across Colombia. Variations in training are observed in the wide ranges of skills completions reported by respondents. These results suggest that graduating medical students in Colombia could benefit from different pedagogic strategies to enhance this knowledge, such as increasing simulation practices, increasing exposure to patients with emergent conditions and critical procedures. The BEC may act as a framework for instructors new to these teaching modalities in addition to potentially providing a baseline standardized curriculum accessible to all medical schools.

Given the availability, lack of subscription fee, low-to-middle income country (LMIC) context specific curriculum as well as the recent translation into Spanish, the BEC may address an educational gap missing in other emergency care courses. Results demonstrate that other courses are occasionally used in medical student training in Colombia. However, several barriers likely exist with these courses. First, they are often have a narrow focus (such as neonatal resuscitation) or are more appropriate only for high-resourced settings [21, 22]. In addition, several, such as ACLS or ATLS are expensive and require that instructors complete a certification process [23]. The BEC is free and open access. It specifically targets emergency care in low resource settings which will be the realistic practice setting for many of these new physicians. In addition, this data demonstrated a dose response, with increasing knowledge and confidence with the higher number of courses completed, supporting the addition of the BEC. Thus, it may be a worthwhile intervention with this learner group.

Knowledge scores and confidence scores measured across all regions of Colombian that have medical schools in Colombia suggest national utility. However, lower knowledge scores in non-accredited schools suggest these students may receive a greater benefit.

Evidence of improvement post-BEC from other countries, specifically in areas where learners showed weaknesses in confidence and knowledge, also offers a compelling argument that the BEC could have a powerful implications in Colombia. For example, a recently published 2017 quasi-experimental study based in Tanzania and Uganda identified that participants of a 5-day BEC training shows a significant increase in emergency care knowledge and confidence at all four study sites [20]. Similarly in Nigeria, post-BEC test scores showed a significant improvement as compared to pre-course (73% vs. 86.5%, $p < 0.001$) [17].

## Limitations

This electronically administered survey had a low response rate, raising the concern for a non-response bias. Low response rates in surveys of healthcare professionals has been well documented in the literature [15]. In addition, even lower response rates amongst physicians have also been noted in LMICs consistent with the response rate in the study [16–18]. However, research has suggested that nonresponse bias is less of a concern amongst the physician population [15, 19–21]. In addition, there were no statistical differences in patient variables across participants removed for incomplete survey and those analyzed in the final cohort. Moreover, the desired sample size for appropriate statistical assessments was reached, supporting validity in the findings.

As with the national distribution of medical students, the majority attended medical school in the Andes region followed by the Caribe region. While the geographic distribution of study participants was statistically different from the national distribution of medical students [13] with an overrepresentation from the Andes and Orinoquía regions and underrepresentation from the Pacífico and Caribe regions (S5 Table in S2 File), geographically stratified analysis demonstrated knowledge and confidence in all regions which would benefit from BEC implementation. Thus, final conclusions remain the same despite potential selection bias.

This survey was distributed online. All medical schools, and therefore medical students, have access to the internet and computers. However, the ability to complete the survey at home, during personal time, may introduce an economic status bias.

While length of emergency medicine rotations was collected, timing was not explored. Knowledge retention, given when this training was completed, may influence results. Multiple choice question assessments are used throughout medical education as a surrogate for overall knowledge with medical students adept at test taking skills. However, with only three

questions asked per topic, one missed question or confusion by a participant greatly reduces the overall score.

In addition, students may perform well at the bedside but struggle with the multiple-choice format. However, this format is utilized in the direct evaluation of BEC evaluation. Future evaluations of clinically relevant and patient centered outcomes pre and post BEC implementation in Colombia will further address this limitation and be a key aspect of monitoring and evaluation.

In Colombia, legislation prevents medical students from performing invasive procedures, such as chest tube placement or intubation, on patients. Thus, the majority, if not all, of pre-graduate skill development is via simulation. Implementation of the BEC at the medical school level in Colombia represents an opportunity to increase early exposure to simulation-based education. ASCOFAME continues to petition the government to change these laws. However, this current legal landscape further informs results. While simulation will never fully replace real-world experience, exercises performed within the BEC would provide an additional opportunity for task training.

## Conclusion

Piloting the BEC in Colombia at central locations offers the potential for a novel implementation of the course amongst a potentially high-impact learner group. Incorporation into medical school curriculum would utilize currently available resources and could represent a sustainable model of implementation. Improved knowledge and confidence in emergency care amongst graduating physicians could theoretically improve the outcomes of patient populations from resource-limited and remote environments, as has been shown in other similar settings [24–26].

Given Colombia's leading medical education system, similar survey results demonstrating BEC appropriateness for senior medical students would be expected across Latin America. The future experience of Colombia in BEC implementation could offer important guidance to the region, leading the way in novel emergency care training.

## Supporting information

**S1 File.**
(PDF)

**S2 File.**
(DOCX)

## Author Contributions

**Conceptualization:** Katelyn Moretti, Adam R. Aluisio, Benjamin Gallo Marin.

**Data curation:** Katelyn Moretti, Adam R. Aluisio.

**Formal analysis:** Katelyn Moretti, Adam R. Aluisio, Benjamin Gallo Marin, Chuan-Jay Jeffrey Chen, Catalina González Marqués, Francesca L. Beaudoin, Melissa Clark, Andrés Patiño, Heidy Carranza, Andres Duarte, Atilio Moreno, Leonar G. Aguiar, Christian Arbelaez.

**Funding acquisition:** Katelyn Moretti, Adam R. Aluisio.

**Investigation:** Katelyn Moretti, Adam R. Aluisio, Benjamin Gallo Marin, Chuan-Jay Jeffrey Chen, Leonar G. Aguiar.

**Methodology:** Katelyn Moretti, Adam R. Aluisio, Leonar G. Aguiar.

**Project administration:** Katelyn Moretti, Benjamin Gallo Marin, Heidy Carranza, Leonar G. Aguiar.

**Resources:** Katelyn Moretti, Leonar G. Aguiar.

**Software:** Katelyn Moretti.

**Supervision:** Katelyn Moretti, Adam R. Aluisio, Leonar G. Aguiar, Christian Arbelaez.

**Validation:** Katelyn Moretti, Adam R. Aluisio.

**Visualization:** Katelyn Moretti, Adam R. Aluisio.

**Writing – original draft:** Katelyn Moretti, Adam R. Aluisio, Benjamin Gallo Marin, Catalina González Marqués, Francesca L. Beaudoin, Leonar G. Aguiar.

**Writing – review & editing:** Katelyn Moretti, Adam R. Aluisio, Benjamin Gallo Marin, Chuan-Jay Jeffrey Chen, Catalina González Marqués, Francesca L. Beaudoin, Melissa Clark, Andrés Patiño, Heidy Carranza, Andres Duarte, Atilio Moreno, Leonar G. Aguiar, Christian Arbelaez.

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
