## [Decision Letter · Decision Letter 0]

5 Aug 2021

PONE-D-21-15138

Confidence and Knowledge in Emergency Management Among Medical Students Across Colombia: A Role for the BEC

PLOS ONE

Dear Authors,

Thank you for submitting your manuscript to PLOS ONE. After careful consideration, we feel that it has merit but does not fully meet PLOS ONE’s publication criteria as it currently stands. Therefore, we invite you to submit a revised version of the manuscript that addresses the points raised during the review process.

We look forward to receiving your revised manuscript.

Kind regards,

Marcel Pikhart

Academic Editor

PLOS ONE

3. We note that Figure 2 in your submission contain [map/satellite] images which may be copyrighted. All PLOS content is published under the Creative Commons Attribution License (CC BY 4.0), which means that the manuscript, images, and Supporting Information files will be freely available online, and any third party is permitted to access, download, copy, distribute, and use these materials in any way, even commercially, with proper attribution. For these reasons, we cannot publish previously copyrighted maps or satellite images created using proprietary data, such as Google software (Google Maps, Street View, and Earth). For more information, see our copyright guidelines: http://journals.plos.org/plosone/s/licenses-and-copyright.

5. Supplemental table 3 is not included in the manuscript.

Additional Editor Comments (if provided):

Reviewers' comments:

Reviewer's Responses to Questions

**Comments to the Author**

1. Is the manuscript technically sound, and do the data support the conclusions?

Reviewer #1: Partly

Reviewer #2: Yes

Reviewer #3: Yes

Reviewer #4: Yes

Reviewer #5: Yes

Reviewer #6: Partly

2. Has the statistical analysis been performed appropriately and rigorously? 

Reviewer #1: Yes

Reviewer #2: Yes

Reviewer #3: Yes

Reviewer #4: Yes

Reviewer #5: Yes

Reviewer #6: No

3. Have the authors made all data underlying the findings in their manuscript fully available?

Reviewer #1: No

Reviewer #2: Yes

Reviewer #3: Yes

Reviewer #4: No

Reviewer #5: Yes

Reviewer #6: Yes

4. Is the manuscript presented in an intelligible fashion and written in standard English?

Reviewer #1: Yes

Reviewer #2: Yes

Reviewer #3: Yes

Reviewer #4: Yes

Reviewer #5: Yes

Reviewer #6: Yes

5. Review Comments to the Author

Reviewer #1: Dear authors,

There are some issues that need to be considered:

1- It is not suitable that the title contains abbreviations. Please write the full form too.

2- Have BEC been implemented and tested for the participants other than medical students? (for example health professionals and practitioners). If so, please mention some of the articles that have utilized this course before in the introduction and compare the results in the discussion.

3- Please state the ethical approval code.

4- How many invitation emails were sent exactly? Was it 46+9? In some parts there are confusions. Also for the samples.

5- Why did you use 100 mm visual analog scales? Mention the rationale please.

6- It is better that you relocate some parts of the methods to the results (e.g. lines 129-132).

7- What is the table in page 18? It does not contain any header and descriptions.

8- Figures and tables do not have high quality and good design. they need to be revised.

9- Why did you insert and refer to supplement tables 4,5 earlier than 1,2?

10- Lines 162-165 should be placed in the “participants and sampling section”.

11- The numerical results should be placed in the results section not the methods.

12- The discussion does not contain enough information. In other words, there is no discussion and interpretation of the results considering the previous research.

13- Overall, the cohesion of the manuscript was not provided and the presentation of work needs to be extensively re-considered.

Reviewer #2: Confidence and Knowledge in Emergency Management Among Medical Students Across Colombia: A Role for the BEC.

Research Article

ID: PONE-D-21-15138

Thank you for asking me to review the above-titled manuscript. The paper topic is interesting, and the authors have presented an interesting study. However, there are a few issues.

Title: BEC should be stated in full, then add "WHO"

Abstract: Knowledge score, add mean±SD, and for confidence score.

Introduction adds more information about BEC of the World Health Organisation, its curriculum, teaching methods, etc. Also, add the URL of the program. Do colleges using it needs permission?

Introduction: State your research question.

Methods: State the number and date of IRB approval.

Please provide more details on the stages of construction of the questionnaire and its validation. Did other published studies have their questionnaire; why the authors did not use a published and validated questionnaire?

Discussion: Could be strengthened. What can this study add compared to other published studies on BEC, WHO?

References- could be improved. The study below is related and should be added.

Broccoli MC, Dixon J, Skarpiak B, Phiri G, Muck AE, Calvello Hynes EJ. Application of the World Health Organization's Basic Emergency Care course in Zambia. Afr J Emerg Med. 2021 Mar;11(1):140-143. doi: 10.1016/j.afjem.2020.09.011. Epub 2020 Oct 17. PMID: 33680735; PMCID: PMC7910165.

Reviewer #3: This is an interesting study, that uses appropriate methodology and statistical analysis. I was gratified to see that concerns I had during reading the work, such as non-response bas, was adequately discussed in the Limitations section of the Discussion. There are minor grammar/punctuation issues; these can be addressed in copy editing.

Reviewer #4: The author has explained the research aim and research process clearly; however, some notes were identified to improve the quality of the paper

In the discussion section of the article, it is better to determine the application of the study findings and to make a more accurate comparison with the study results

Please add some new references in the last section

Reviewer #5: This is an interesting report that describes the deliver of the WHO's Basic Emergency Care training program for the first time in Latin America, and highlights the benefits, of an early stage intervention, of that training for new graduates, as well being considered for increased inclusion in medical curricula.

The report provides evidence, as well as a useful discussion about the merits of early training emergency care and in particular using the BEC training program in helping new graduates and medical students gain valuable experience, via a low cost training program for providers with limited resources.

The results demonstrate that the BEC successfully improved the knowledge and confidence of new graduates and medical students in being able to provide emergency care, and that the course could be considered by other nations, as a way to bolster healthcare services, cheaply and efficiently to save lives.

Reviewer #6: This is a cross-sectional study assessing the knowledge and self-reported confidence of last-year (4th-year) medical students in emergency medicine areas corresponding to The Basic Emergency Care course (BEC) developed by the World Health Organization to evaluate its potential utility at the medical student level in Colombia. As the medical graduates without further residency training serve vulnerable populations in Colombia, BEC was hypothesised to be beneficial in improving the patient outcomes if the students’ baseline knowledge is below the a priori value of 75%. Additionally, because BEC was initially designed for and previously piloted on frontline providers without formal EM training, this study may create an impact by encouraging other Latin American countries or LMICs to consider BEC as an alternative in medical student groups.

Overall, this manuscript explains the methodology, especially, the creation and the validation of the questionnaire well. It is linguistically well written and structured. However, in addition to minor mistakes, there are areas that should be improved / more clearly explained in all sections. Therefore, the manuscript needs improvement and cannot be accepted in this current form. Here are my specific comments:

Major comments

Title and abstract:

(1) The title and abstract introduce the study adequately.

Introduction:

(2) I would suggest revising the introduction, especially the first paragraph, as it can be divided into several paragraphs with singular key ideas and deeper explanations, and the whole introduction can be rearranged accordingly, following the problem/gap/hook technique more, to improve the manuscript.

(3) (Also see comments on discussion) There are other potential courses that can be implemented in the same setting to advance medical graduates' preparedness to practice. How the authors decided to assess the utility of BEC over others is not clear. As most educational interventions (versus no intervention) could theoretically and empirically increase trainee knowledge and skills, I would advise explaining the rationale behind preferring BEC in the first place more explicitly, comparing it with other potential courses in all relevant aspects, including, for example, scope, feasibility and accessibility.

(4) While the manuscript does not explicitly state objectives or research questions, what the study aims to investigate can be inferred from the last paragraph of the introduction. The authors should consider further clarifying the research question(s)/hypotheses.

Methodology:

(5) The authors described the study design and setting adequately. The data collection method is congruent with what the study evaluates.

(6) The participants, questionnaire design and timeframe were reported adequately. I also find the description of pilot-testing, validation and data security efforts valuable. However, I would suggest explaining the need to create and validate a new questionnaire. Was there any pre-validated alternative in the literature? If any, why did the authors deem the alternative(s) inadequate for the research aim?

(7) The study size was defined as required. However, I would advise explaining the rationale behind the choice of statistical tests more clearly. This discussion should include, for example, the data and variable types, and whether the data is distributed normally, how the use of parametric or non-parametric tests are decided. Additionally, the authors should explicitly state the p-value cut-off deemed statistically significant for this study (e.g., <0.05 vs <0.01).

(8) I could not find the questionnaire among the documents provided. The original questionnaire in Spanish (or its English translation) should be added as an appendix. 

Results:

(9) Participants and descriptive data (including Figure 1-2) were reported adequately.

(10) Where p-values are stated, the test used should be stated in the text (as well as in tables, see next comment)

(11) The tables should explicitly state the statistical test(s) used. Also, in supplemental table 1 (Starting from line 200), it is not clear which two means are compared to reach the p values. If the mean knowledge (and confidence?) levels were compared to the a priori maximum score of 75%, this should be stated clearly and rationalized statistically in the methodology section. If other two means (e.g., the mean knowledge of course takers and non-takers) were compared, the table should be revised as such.

Discussion:

(12) The key results were summarised adequately. However, overall, I find the discussion a little superficial. The study design is mostly based on a priori or theoretical assumptions evaluating the circumstances in which BEC could be beneficial rather than implementing the BEC actually. Although this is fine, I think, the discussion is an opportunity to compare and contrast the findings reached as a consequence of these assumptions with studies where BEC was implemented or other empirical studies regarding other variables.

(13) The results sections include quite a lot of data in form of text and tables. However, the discussion did not explain these results or compare and contrast them with the literature. The discussion should match the results or alternatively, the irrelevant results can be omitted. For example, the impact of previous courses, EM rotation times, gender, confidence/self-efficacy vs the number of task performances, being trained in an accredited centre, etc. on knowledge and confidence level in similar/other settings could be discussed.

(14) As the authors mentioned previously that the undergraduate emergency medicine curriculum is not standardised nationally, it may be worth reflecting on this as the reason for wide ranges in skills completions in the second paragraph of the discussion. Additionally, BEC potentially can act as a framework in standardisation efforts.

(15) In the third paragraph, the authors explained how BEC could benefit in the given setting. However, a knowledge/confidence gap can theoretically be filled by more than one educational alternatives, and without discussing the alternatives explicitly (See comments on introduction), the conclusions drawn on the utility and feasibility of BEC seems less objective. The authors should consider discussing other international EM courses/other solutions that can be potentially applied in the same setting, if any, comparing it with BEC in terms of advantages and disadvantages and explaining why BEC was preferred over the others.

(16) In the fourth paragraph, the authors compared local courses with BEC. If there are published works, they should be cited.

(17) In the introduction, multiple studies evaluating the results of BEC in other LMIC contexts were cited. Also, other LMICs might have implemented other courses/solutions for their educational problems, which can offer alternative solutions to the Colombian context, too. These should be discussed in the discussion in detail.

(18) The limitations were discussed extensively. As the questionnaire was conducted online, I would suggest considering access to the internet as a factor in response rate and distribution.

(19) The inadequacies in discussion make the conclusions seem overconfident. A deeper discussion and a more succinct and toned-down conclusion section would make conclusions appear more congruent with the rest of the study.

Funding

(20) The funding information was shared transparently.

Minor Comments

(21) The authors explained BEC fairly well in the introduction. However, this section missed some key information, including how BEC is performed. (For example, in the questionnaire design, it was mentioned that BEC uses low technology simulation to complete task training.) I would suggest elaborating on BEC in a separate paragraph in the introduction, giving all necessary information.

(22) It can be valuable to add approximately how much time completing the questionnaire takes to interpret the response rate.

(23) The confidence level was assessed by 13 questions. However, it is also mentioned that 3 questions in each of the 4 categories were asked. This creates confusion about whether 13 was a typo. Otherwise, defining the final question could clear the confusion.

(24) Though the number of invitees and participants were specified, the authors should consider giving the response rate as a percentage.

(25) I think there are mistakes regarding table names and numbers and also mismatches between text and corresponding tables. The order of tables should be checked. In general, the data regarding the table in text should precede the table.

(26) p values that are now reported as 0 should be replaced by <0.001

(27) I understand that medical school is four years in Colombia from the explanation of table 2. As there are various models available (e.g., six-year, seven-year, etc.), this should be stated in the introduction.

(28) Emergency Medicine "Areas" should be added to the name of Table 2, as the name suggests that BEC had already been implemented in the current form.

(29) The figure 1 might be lacking arrows, please check.

6. PLOS authors have the option to publish the peer review history of their article (what does this mean?). If published, this will include your full peer review and any attached files.

Reviewer #1: No

Reviewer #2: **Yes: **PROFESSOR SAMY A AZER

Reviewer #3: No

Reviewer #4: No

Reviewer #5: No

Reviewer #6: No

---

## [Author Response · Author response to Decision Letter 0]

2 Nov 2021

October 25, 2021

Dear Editor, 

Response and resubmission: PONE-D-21-15138

Thank you for your correspondence on August 05, 2021. We appreciate the chance to amend and resubmit our manuscript, “Confidence and Knowledge in Emergency Management Among Medical Students Across Colombia: A Role for the WHO Basic Emergency Care Course” to the PLOSone Journal for consideration for publication as an original article. Please note that the manuscript has been changed to address the points raised by the peer reviewers. Each point is addressed below with the corresponding alterations in the manuscript explained. 

As previously noted, we believe the reported data and associated manuscript fit well with your Journal’s mission to communicate how systems and delivery of emergency care affect populations worldwide. Our manuscript is the first to suggest that there may be a space for a sustainable and innovative enhancement to Colombia’s medical school curricula through the addition of the World Health Organization’s (WHO) Basic Emergency Course (BEC). Specifically, we examine baseline knowledge and confidence among senior medical students across Colombia in the content presented in the WHO Basic Emergency Care course. We believe that our manuscript will be of great interest to your diverse readership.

Thank you very much for your time and assistance. 

Sincerely, 

Katelyn Moretti, MD

Warren Alpert School of Medicine, Brown University

Department of Emergency Medicine

55 Claverick Street, Room 274

Providence, Rhode Island 02903, USA

Telephone: +1(401)444-5826

Email: Katelyn_moretti@brown.edu

Responses to the Editors below:

Reviewer #1: Dear authors,

There are some issues that need to be considered:

1- It is not suitable that the title contains abbreviations. Please write the full form too.

• Thank you for this feedback. The title has been updated. 

2- Have BEC been implemented and tested for the participants other than medical students? (for example health professionals and practitioners). If so, please mention some of the articles that have utilized this course before in the introduction and compare the results in the discussion.

• Thank for this feedback. The introduction was modified to include: 

“In 2015, the WHO successfully piloted the BEC course with frontline providers including medical officers, nurses, and nursing assistants in Uganda, United Republic of Tanzania, and Zambia6. Post-test knowledge scores and confidence levels were significantly improved after implementation.5,7” (Lines 97-99)

• The discussion was modified to include:

“There was no individual group analyzed in this study that had a mean score above the a priori maximum score of 75%. In fact, knowledge scores were similar to the pretest scores of previous BEC learner groups which included frontline providers such as medical officers and nurses from Tanzania, Uganda and Zambia5. Similarly, confidence scores for the management of patients needing emergent care for various life-threatening situations were low, suggesting an opportunity for curricular enhancements.” (Lines 327-333)

3- Please state the ethical approval code.

• Thank you. The ethical approval code has been added:

 “This cross-sectional study was conducted among Colombian medical students in their last year of medical school and approved through the institutional review board at Pontifical Javeriana University and the Hospital Universitario San Ignacio (CA/004-2019)” (Lines 119-121). 

4- How many invitation emails were sent exactly? Was it 46+9? In some parts there are confusions. Also for the samples.

• Thank you for this feedback. The following section was edited for clarity.

“Invitations to participate were successfully sent to 65% of the Colombian medical student body. Given the reliance on intermediates for survey distribution, the study population was defined as the student population from medical schools with at least one student accessing the survey. This included the student body of 36 medical schools across Colombia encompassing 4,166 students. Of this initial sample, 714 students consented to participate; 468 completed the knowledge and confidence aspects of the survey and were included in analysis (Fig 1).” (Lines 229-240)

5- Why did you use 100 mm visual analog scales? Mention the rationale please.

• Thank you for this feedback. The following rationale and citation has been included: 

“Confidence level was assessed via 13 questions using 100 mm visual analog scales. Questions addressed confidence specific to managing patients with dyspnea, shock, trauma, altered mental status (AMS), with three items per category. Visual analogue scales have previously been validated for the measurement of self-efficacy in resuscitation.9” (Lines 144-149)

6- It is better that you relocate some parts of the methods to the results (e.g. lines 129-132).

• The following section was moved to methods:

“A 75% score was defined, a priori, as the maximum score for course utility below which, potential knowledge gains would warrant implementation. This is consistent with the defined passing score of the knowledge test provided in the BEC5.” (Lines 274-276)

7- What is the table in page 18? It does not contain any header and descriptions.

• We only see references on page 18. However, all tables have been reviewed, edited and renumbered as necessary. All should have descriptors. 

8- Figures and tables do not have high quality and good design. they need to be revised.

• The tables have been edited and we feel are greatly improved. The author team thanks you for this feedback. 

9- Why did you insert and refer to supplement tables 4,5 earlier than 1,2?

• Thank you for this insightful feedback. The numbering of the supplemental tables has been updated to reflect their referral in the manuscript. 

10- Lines 162-165 should be placed in the “participants and sampling section”.

• Thank you for this feedback. Lines 162-165 explain the rationale for how a completed survey was defined. This section has been clarified in the following manner:

“Items in both the knowledge and confidence section may have been missing based on participants stopping the survey early (failure to complete) or, because the participant did not know the answer and skipped that specific question (failure to respond). Thus, 

completion of the last question of the knowledge or confidence section was analyzed to distinguish between failure to complete versus failure to respond. Based on this standardized approach, a complete knowledge score was defined as 14 or more questions answered and a complete confidence score was defined as greater than 10 items answered (S1 Table and S2 Table).” (Lines 177 – 184)

11- The numerical results should be placed in the results section not the methods.

• Thank you for this feedback. Several sections have been moved to results including

o The number of invitations sent by partnering organizations. 

o Pilot data

12- The discussion does not contain enough information. In other words, there is no discussion and interpretation of the results considering the previous research.

• Thank you for this feedback. The discussion has been expanded with the following section:

“As of the date of submission, this is the first attempt to identify medical students as a learner group for the WHO, besides one study that included 2 medical students in Nigeria.16 Moreover, of all the studies examining emergency care in LMICs, this is the first to break down results by provider gender, age, and number of previous training courses. Interestingly, a number of trends did emerge from the results. Confidence in providing emergency care was higher among male than female students, correlating with studies in other settings that show male medical students to appear more confident than female students in clinical encounters.17 Meanwhile, there was no statistical difference in knowledge scores between genders. Similar to other studies, both confidence and knowledge increased with the number of training courses that students had previously taken.1,2 This suggests that implementing the BEC would potentially also increase the knowledge and confidence of medical students who will go on to provide emergency medical care to the vulnerable, underserved, often rural populations they will serve during their year of social service as it has in other settings.6,18,19 implementation of the BEC at the medical school level could increase the knowledge and confidence of medical students who will go on to provide emergency medical care to the vulnerable, underserved, often rural populations during their year of social service.6,19,20” (Lines 334 – 346)

13- Overall, the cohesion of the manuscript was not provided and the presentation of work needs to be extensively re-considered.

• Thank you for all of your valuable insights. With your and other reviewer feedback, several large edits in both organization and content have been made which I feel greatly enhances the manuscript. 

Reviewer #2: Confidence and Knowledge in Emergency Management Among Medical Students Across Colombia: A Role for the BEC.

Research Article

ID: PONE-D-21-15138

Thank you for asking me to review the above-titled manuscript. The paper topic is interesting, and the authors have presented an interesting study. However, there are a few issues.

Title: BEC should be stated in full, then add "WHO"

• Thank you for this feedback. These changes have been made.

Abstract: Knowledge score, add mean±SD, and for confidence score.

• Thank you. Standard deviations have been added to text. 

Introduction adds more information about BEC of the World Health Organisation, its curriculum, teaching methods, etc. Also, add the URL of the program. Do colleges using it needs permission?

• Thank you for this feedback. Additional information about the BEC and its format, teaching methods as well as the URL for the program has been added. Clarification of its free, open-access status has also been added: 

“The BEC is delivered as five modules (excluding an Introduction module). A Participant Workbook is given to learners, which includes the content of the course along with exercises that include free-response case-based questions and four-option multiple choice questions. The Participant Workbook also includes a Skills Station section with a checklist-based approached to the skills learners must demonstrate. In addition, supplemental mobile applications that provide interactive case-based training on trauma ABCDEs (Airway, Breathing, Circulation, Disability, Exposure) are also available. Instructors are provided with an annotated Facilitator Guide along with digital slides for each module that follow the sequence of the Participant Workbook. The course is both free and open-access and can be accessed through the WHO website7.” (Lines 101 – 109) 

Introduction: State your research question.

• Thank you for this feedback. The following has been added to the introduction:

“This study assessed the baseline knowledge and confidence in emergency care that is specifically taught in BEC curriculum amongst graduating medical students to answer the question: does the BEC offer an appropriate supplemental curriculum for medical students across Colombia.” (Lines 113-116) 

Methods: State the number and date of IRB approval.

• The number and date have been added. 

o (FM-CIE-0279-19, 05/13/2019). (Line 121) 

Please provide more details on the stages of construction of the questionnaire and its validation. Did other published studies have their questionnaire; why the authors did not use a published and validated questionnaire?

• Thank you for this feedback. The following sections, addressing your concerns were added:

“The survey was written in Spanish and designed to assess students’ baseline knowledge and confidence in the BEC content domains. While the course was open-access at the time of this study, formal knowledge assessments were not yet available. While there were some review questions as the end of course sections, these were not enough for a full evaluation. Therefore, a new evaluation tool, with questions targeted to a similar knowledge level as those provided in the course, was created.”(Lines 132 – 137) 

Discussion: Could be strengthened. What can this study add compared to other published studies on BEC, WHO?

References- could be improved. The study below is related and should be added.

Broccoli MC, Dixon J, Skarpiak B, Phiri G, Muck AE, Calvello Hynes EJ. Application of the World Health Organization's Basic Emergency Care course in Zambia. Afr J Emerg Med. 2021 Mar;11(1):140-143. doi: 10.1016/j.afjem.2020.09.011. Epub 2020 Oct 17. PMID: 33680735; PMCID: PMC7910165.

• Thank you for this feedback the discussion has been updated and lengthened to compare to previous studies as suggested above. In addition, the following citations, including the citation kindly suggested above, have been added

16. Olufadeji A, Usoro A, Akubueze CE, et al. Results from the implementation of the World Health Organization Basic Emergency Care Course in Lagos, Nigeria. Afr J Emerg Med 2021;11(2):231–6. 

17. Blanch DC, Hall JA, Roter DL, Frankel RM. Medical student gender and issues of confidence. Patient Educ Couns 2008;72(3):374–81. 

18. Broccoli MC, Dixon J, Skarpiak B, Phiri G, Muck AE, Calvello Hynes EJ. Application of the World Health Organization’s Basic Emergency Care course in Zambia. Afr J Emerg Med 2021;11(1):140–3. 

19. Kivlehan SM, Dixon J, Kalanzi J, et al. Strengthening emergency care knowledge and skills in Uganda and Tanzania with the WHO-ICRC Basic Emergency Care Course. Emerg Med J 2021;38(8):636–42. 

20. Qureshi F, Hafeez A, Zafar S, Mohamud BK, Southall DP. Evidence for improvement in the quality of care given during emergencies in pregnancy, infancy and childhood following training in life-saving skills: a postal survey. J Pak Med Assoc 2009;59(1):5. 

21. Effect of Emergency Medical Technician Certification for All... : Journal of Trauma and Acute Care Surgery [Internet]. [cited 2021 Oct 4];Available from: https://journals.lww.com/jtrauma/pages/articleviewer.aspx?year=2007&issue=10000&article=00030&type=abstract

22. Improvements in prehospital trauma care in an African country with no formal emergency medical services. [Internet]. Epistemonikos. [cited 2021 Oct 4];Available from: https://www.epistemonikos.org/it/documents/2ef8d51c78642464d5a89537e3e633599184071

Reviewer #3: This is an interesting study, that uses appropriate methodology and statistical analysis. I was gratified to see that concerns I had during reading the work, such as non-response bas, was adequately discussed in the Limitations section of the Discussion. There are minor grammar/punctuation issues; these can be addressed in copy editing.

• Thank you for the kind feedback!

Reviewer #4: The author has explained the research aim and research process clearly; however, some notes were identified to improve the quality of the paper

In the discussion section of the article, it is better to determine the application of the study findings and to make a more accurate comparison with the study results

Please add some new references in the last section

• Thank you for this feedback. Additional citations, as detailed above, have been added. 

Reviewer #5: This is an interesting report that describes the deliver of the WHO's Basic Emergency Care training program for the first time in Latin America, and highlights the benefits, of an early stage intervention, of that training for new graduates, as well being 

considered for increased inclusion in medical curricula.

The report provides evidence, as well as a useful discussion about the merits of early training emergency care and in particular using the BEC training program in helping new graduates and medical students gain valuable experience, via a low cost training program for providers with limited resources.

The results demonstrate that the BEC successfully improved the knowledge and confidence of new graduates and medical students in being able to provide emergency care, and that the course could be considered by other nations, as a way to bolster healthcare services, cheaply and efficiently to save lives.

• Thank you for your review. 

Reviewer #6: This is a cross-sectional study assessing the knowledge and self-reported confidence of last-year (4th-year) medical students in emergency medicine areas corresponding to The Basic Emergency Care course (BEC) developed by the World Health Organization to evaluate its potential utility at the medical student level in Colombia. As the medical graduates without further residency training serve vulnerable populations in Colombia, BEC was hypothesised to be beneficial in improving the patient outcomes if the students’ baseline knowledge is below the a priori value of 75%. Additionally, because BEC was initially designed for and previously piloted on frontline providers without formal EM training, this study may create an impact by encouraging other Latin American countries or LMICs to consider BEC as an alternative in medical student groups.

Overall, this manuscript explains the methodology, especially, the creation and the validation of the questionnaire well. It is linguistically well written and structured. However, in addition to minor mistakes, there are areas that should be improved / more clearly explained in all sections. Therefore, the manuscript needs improvement and cannot be accepted in this current form. Here are my specific comments:

Major comments

Title and abstract:

(1) The title and abstract introduce the study adequately.

Introduction:

(2) I would suggest revising the introduction, especially the first paragraph, as it can be divided into several paragraphs with singular key ideas and deeper explanations, and the whole introduction can be rearranged accordingly, following the problem/gap/hook technique more, to improve the manuscript.

• Thank you for this feedback. The introduction has been significantly restructured with additional explanation of the BEC and its merits. In addition, the knowledge gap of medical students knowledge and why it is important, has been expanded upon. 

(3) (Also see comments on discussion) There are other potential courses that can be implemented in the same setting to advance medical graduates' preparedness to practice. How the authors decided to assess the utility of BEC over others is not clear. As most educational interventions (versus no intervention) could theoretically and empirically increase trainee knowledge and skills, I would advise explaining the rationale behind preferring BEC in the first place more explicitly, comparing it with other potential courses in all relevant aspects, including, for example, scope, feasibility and accessibility.

• Thank you for this feedback and suggestion. The following section has been added:

“Results demonstrate that other courses are occasionally used in medical student training in Colombia. However, several barriers likely exist with these courses. First, they are often have a narrow focus (such as neonatal resuscitation) or are more appropriate only for high-resourced settings.20,21 In addition, several, such as ACLS or ATLS are expensive and require that instructors complete a certification process.22 The BEC is free and open access. It specifically targets emergency care in low resource setting s which will be the realistic practice setting for many of these new physicians. Thus, it may be a worthwhile intervention with this learner group.”(Lines 357 – 366) 

(4) While the manuscript does not explicitly state objectives or research questions, what the study aims to investigate can be inferred from the last paragraph of the introduction. The authors should consider further clarifying the research question(s)/hypotheses.

• Thank you for this feedback. The following has been added:

“The BEC has never been implemented specifically for medical students nor in Latin America; therefore, its pertinence and associated impacts for this learner group is unclear. This study assessed the baseline knowledge and confidence in emergency care that is specifically taught in BEC curriculum amongst graduating medical students to answer the question: does the BEC offer an appropriate supplemental curriculum for medical students across Colombia.” (Lines 110 – 116) 

Methodology:

(5) The authors described the study design and setting adequately. The data collection method is congruent with what the study evaluates.

• Thank you for this feedback.

(6) The participants, questionnaire design and timeframe were reported adequately. I also find the description of pilot-testing, validation and data security efforts valuable. However, I would suggest explaining the need to create and validate a new questionnaire. Was there any pre-validated alternative in the literature? If any, why did the authors deem the alternative(s) inadequate for the research aim?

• Thank you for this feedback. As detailed above, a section has been added explaining the rationale for a new assessment tool. 

(7) The study size was defined as required. However, I would advise explaining the rationale behind the choice of statistical tests more clearly. This discussion should include, for example, the data and variable types, and whether the data is distributed normally, how the use of parametric or non-parametric tests are decided. Additionally, the authors should explicitly state the p-value cut-off deemed statistically significant for this study (e.g., <0.05 vs <0.01).

• Thank you for this feedback. The following has been added:

“Data were examined graphically and found to be normally distributed. Parametric comparative assessments were used (two-tailed t-test and ANOVA) to evaluate for significant differences between stratified groups of interest. The Benjamini-Hochberg method for controlling for the false discovery rate14 was used to determine the statistical significance level of P �0.014. The method was applied to P values for comparisons of mean knowledge scores and again for mean confidence scores. The more conservative P value was then selected and used consistently for all values. “ (Lines 199 – 205) 

(8) I could not find the questionnaire among the documents provided. The original questionnaire in Spanish (or its English translation) should be added as an appendix. 

• The questionnaire has been added. 

Results:

(9) Participants and descriptive data (including Figure 1-2) were reported adequately.

• Thank you. 

(10) Where p-values are stated, the test used should be stated in the text (as well as in tables, see next comment)

• Thank you. We have now done this. 

(11) The tables should explicitly state the statistical test(s) used. Also, in supplemental table 1 (Starting from line 200), it is not clear which two means are compared to reach the p values. If the mean knowledge (and confidence?) levels were compared to the a priori maximum score of 75%, this should be stated clearly and rationalized statistically in the methodology section. If other two means (e.g., the mean knowledge of course takers and non-takers) were compared, the table should be revised as such.

• Thank you for the feedback. Statistical tests have been added to table descriptors. Supplemental table 1 is now named supplemental table 3. Mean knowledge and confidence for those 

completing a course was compared to the overall mean knowledge or mean knowledge score. This has been added to the table descriptor. 

• Added to Methods:

“Knowledge scores and confidence scores for students who completed previous emergency care courses were compared to the overall knowledge score or confidence score of the study sample using two-sided T-Test.”

• Added to Results:

“Students who had completed BLS, ACLS, Pediatric Advanced Life Support, Neonatal Advanced Life Support or Atención Integral a las Enfermedades Prevalentes en la Infancia scored higher on the knowledge assessment as compared to the overall cohort. Those that had completed BLS or ACLS also demonstrated higher confidence levels.”

Discussion:

(12) The key results were summarised adequately. However, overall, I find the discussion a little superficial. The study design is mostly based on a priori or theoretical assumptions evaluating the circumstances in which BEC could be beneficial rather than implementing the BEC actually. Although this is fine, I think, the discussion is an opportunity to compare and contrast the findings reached as a consequence of these assumptions with studies where BEC was implemented or other empirical studies regarding other variables.

• Thank you for this feedback. The following section has been added to the discussion:

“This study is the first step to gauge appropriateness of the BEC in Colombia, a new setting, amongst a novel learner group. Based on the findings of studies that examine the participant knowledge and confidence of emergency care in settings outside of Colombia and results from this study, we hypothesize that in Colombia, a similar positive impact would be observed. For example, a recently published 2017 quasi-experimental study based in Tanzania and Uganda identified that participants of a 5-day BEC training shows a significant increase in emergency care knowledge and confidence at all four study sites.19 Similarly in Nigeria, post-BEC test scores showed a significant improvement as compared to pre-course (73% vs. 86.5%, p < 0.001).16 Evidence of improvement post-BEC from other countries, specifically in areas where learners showed weaknesses in confidence and knowledge, offers a compelling argument that the BEC could have a powerful implications in Colombia.” (Lines 419 – 429) 

(13) The results sections include quite a lot of data in form of text and tables. However, the discussion did not explain these results or compare and contrast them with the literature. The discussion should match the results or alternatively, the irrelevant results can be omitted. For example, the impact of previous courses, EM rotation times, gender, confidence/self-efficacy vs the number of task performances, being trained in an accredited centre, etc. on knowledge and confidence level in similar/other settings could be discussed.

• Thank you for this feedback. As detailed above, the discussion has been lengthened to include a more thorough discussion of results. 

(14) As the authors mentioned previously that the undergraduate emergency medicine curriculum is not standardised nationally, it may be worth reflecting on this as the 

reason for wide ranges in skills completions in the second paragraph of the discussion. Additionally, BEC potentially can act as a framework in standardisation efforts.

• Thank you for this feedback. The discussion has been lengthened to strengthen this point.

(15) In the third paragraph, the authors explained how BEC could benefit in the given setting. However, a knowledge/confidence gap can theoretically be filled by more than one educational alternatives, and without discussing the alternatives explicitly (See comments on introduction), the conclusions drawn on the utility and feasibility of BEC seems less objective. The authors should consider discussing other international EM courses/other solutions that can be potentially applied in the same setting, if any, comparing it with BEC in terms of advantages and disadvantages and explaining why BEC was preferred over the others.

• Thank you for this feedback. The limitations of other EM courses has been added to the discussion. 

(16) In the fourth paragraph, the authors compared local courses with BEC. If there are published works, they should be cited.

• Citations have been added. 

(17) In the introduction, multiple studies evaluating the results of BEC in other LMIC contexts were cited. Also, other LMICs might have implemented other courses/solutions for their educational problems, which can offer alternative solutions to the Colombian context, too. These should be discussed in the discussion in detail.

• Thank you for this feedback. As detailed above, a more in-depth discussion of the results of these studies have been added. 

(18) The limitations were discussed extensively. As the questionnaire was conducted online, I would suggest considering access to the internet as a factor in response rate and distribution.

• Thank you for this insight. The following has been added:

“This survey was distributed online. All medical schools, and therefore medical students, have access to the internet and computers. However, the ability to complete the survey at home, during personal time may introduce an economic status bias.” (Lines 386 – 388) 

(19) The inadequacies in discussion make the conclusions seem overconfident. A deeper discussion and a more succinct and toned-down conclusion section would make conclusions appear more congruent with the rest of the study.

Thank you for this feedback. The discussion has been significantly deepened in response the thoughtful feedback from all reviewers. 

Funding

(20) The funding information was shared transparently.

Minor Comments

(21) The authors explained BEC fairly well in the introduction. However, this section 

 missed some key information, including how BEC is performed. (For example, in the questionnaire design, it was mentioned that BEC uses low technology simulation to complete task training.) I would suggest elaborating on BEC in a separate paragraph in the introduction, giving all necessary information.

• Thank you. As detailed above, a paragraph further explaining the BEC has been added. 

 

(22) It can be valuable to add approximately how much time completing the questionnaire takes to interpret the response rate.

• Thank you for this feedback. The following has been added to results:

“The median time for completion was 17.5 minutes (IQR 13.1-24.0 minutes).” (Line 224)

(23) The confidence level was assessed by 13 questions. However, it is also mentioned that 3 questions in each of the 4 categories were asked. This creates confusion about whether 13 was a typo. Otherwise, defining the final question could clear the confusion.

• Thank you. The following has been added to clarify:

“Confidence level was assessed via 13 questions using 100 mm visual analog scales. Questions addressed confidence specific to managing patients with dyspnea, shock, trauma, altered mental status (AMS), with three items per category for a total of 12 questions. A final question asked for confidence in the management of a critical patient in general. Visual analogue scales were selected as they have previously been validated for the measurement of self-efficacy in resuscitation.10” (Lines 144 – 152) 

(24) Though the number of invitees and participants were specified, the authors should consider giving the response rate as a percentage.

• The response rate as a percentage has been added. 

(25) I think there are mistakes regarding table names and numbers and also mismatches between text and corresponding tables. The order of tables should be checked. In general, the data regarding the table in text should precede the table.

• Thank you. All tables and numbers have been updated. 

(26) p values that are now reported as 0 should be replaced by <0.001

• All have been updated. 

(27) I understand that medical school is four years in Colombia from the explanation of table 2. As there are various models available (e.g., six-year, seven-year, etc.), this should be stated in the introduction.

• There are no alternates. All medical school curriculum is completed in four years. 

(28) Emergency Medicine "Areas" should be added to the name of Table 2, as the name suggests that BEC had already been implemented in the current form.

• This has been updated. 

(29) The figure 1 might be lacking arrows, please check.

• Arrows have been added. 

6. PLOS authors have the option to publish the peer review history of their article (what does this mean?). If published, this will include your full peer review and any attached files.

Do you want your identity to be public for this peer review? For information about this choice, including consent withdrawal, please see our Privacy Policy.

Reviewer #1: No

Reviewer #2: Yes: PROFESSOR SAMY A AZER

Reviewer #3: No

Reviewer #4: No

Reviewer #5: No

Reviewer #6: No

---

## [Decision Letter · Decision Letter 1]

21 Dec 2021

Confidence and Knowledge in Emergency Management Among Medical Students Across Colombia: A Role for the WHO Basic Emergency Care Course

PONE-D-21-15138R1

Dear Authors,

We’re pleased to inform you that your manuscript has been judged scientifically suitable for publication and will be formally accepted for publication once it meets all outstanding technical requirements.

Kind regards,

Marcel Pikhart

Academic Editor

PLOS ONE

Additional Editor Comments (optional):

Reviewers' comments:

Reviewer's Responses to Questions

**Comments to the Author**

1. If the authors have adequately addressed your comments raised in a previous round of review and you feel that this manuscript is now acceptable for publication, you may indicate that here to bypass the “Comments to the Author” section, enter your conflict of interest statement in the “Confidential to Editor” section, and submit your "Accept" recommendation.

Reviewer #6: All comments have been addressed

2. Is the manuscript technically sound, and do the data support the conclusions?

Reviewer #6: Yes

3. Has the statistical analysis been performed appropriately and rigorously? 

Reviewer #6: Yes

4. Have the authors made all data underlying the findings in their manuscript fully available?

Reviewer #6: Yes

5. Is the manuscript presented in an intelligible fashion and written in standard English?

Reviewer #6: Yes

6. Review Comments to the Author

Reviewer #6: Dear authors,

Thanks for addressing the comments. I find this version of the manuscript greatly improved.

My only comment is about the conclusion section. The conclusion includes three long paragraphs with overlapping ideas [For example: "In addition to being the first BEC-related published study in Latin America, this is the first study looking specifically at medical students as potential trainees for the course." (Lines 410-411) and "This study is the first step to gauge appropriateness of the BEC in Colombia, a new setting, amongst a novel learner group." (Lines 420-421)]. In general, now that the rest of the study supports the conclusion more, the authors can express the summary of the study with more concise and clear sentences. Overall, there is no need to repeat previous sentiments fully to re-convince readers; only central ideas should be summarized shortly. I believe the authors can rephrase the conclusion effectively in two paragraphs.

The second paragraph of the conclusion introduces a new argument (Lines 421-428), which should normally take place in the discussion rather than the conclusion. I would recommend shifting the detailed explanation of ideas and comparison with other studies to the discussion section and summarizing central ideas based on this discussion in the conclusion section. In other words, the discussion should include the details supporting the conclusion. Also, there is a duplication of ideas in terms of BEC's potential positive impact in Columbia (Lines 422-424 and Lines 428-430).

In the third paragraph, I do not know if the use of the future tense is deliberate (For example: "... the medical school curricula WILL BE prospectively evaluated for effectiveness and sustainability ... "), but it sounds like the authors have already planned/started exploring mentioned aspects of BEC. However, in general, sticking to the use of "should" may be better as the manuscript also illuminates what the other researchers can evaluate in the future based on the study findings [Line 431 - 435 (the second one)].

In sum, please consider adjusting the conclusion section as a whole in light of the new version of the manuscript one more time. Except for it, the manuscript now appears clear, organized, and congruent to me.

7. PLOS authors have the option to publish the peer review history of their article (what does this mean?). If published, this will include your full peer review and any attached files.

Reviewer #6: No

---

## [Editor Report · Acceptance letter]

11 Jan 2022

PONE-D-21-15138R1 

Confidence and Knowledge in Emergency Management Among Medical Students Across Colombia: A Role for the WHO Basic Emergency Care Course 

Dear Dr. Moretti:

I'm pleased to inform you that your manuscript has been deemed suitable for publication in PLOS ONE. Congratulations! Your manuscript is now with our production department. 

Kind regards, 

on behalf of

Dr. Marcel Pikhart 

Academic Editor

PLOS ONE